# In-Situ Observations of Microscale Ductility in a Quasi-Brittle Bulk Scale Epoxy

**DOI:** 10.3390/polym12112581

**Published:** 2020-11-03

**Authors:** Olivier Verschatse, Lode Daelemans, Wim Van Paepegem, Karen De Clerck

**Affiliations:** Department of Materials, Textiles and Chemical Engineering (MaTCh), Ghent University, Technologiepark 70A, B-9052 Zwijnaarde, Belgium; olivier.verschatse@ugent.be (O.V.); Lode.Daelemans@ugent.be (L.D.); Wim.VanPaepegem@ugent.be (W.V.P.)

**Keywords:** Scanning Electron Microscopy (SEM), micromechanical testing, microscale, yielding, plasticity

## Abstract

Fiber reinforced composite materials are typically comprised of two phases, i.e., the reinforcing fibers and a surrounding matrix. At a high volume fraction of reinforcing fibers, the matrix is confined to a microscale region in between the fibers (1–200 µm). Although these regions are interconnected, their behavior is likely dominated by their micro-scale. Nevertheless, the characterization of the matrix material (without reinforcing fibers) is usually performed on macroscopic bulk specimens and little is known about the micro-mechanical behavior of polymer matrix materials. Here, we show that the microscale behavior of an epoxy resin typically used in composite production is clearly different from its macroscale behavior. Microscale polymer specimens were produced by drawing microfibers from vitrifying epoxy resin. After curing, tensile tests were performed on a large set of pure epoxy microfiber specimens with diameters ranging from 30 to 400 µm. An extreme ductility was observed for microscale epoxy specimens, while bulk scale epoxy specimens showed brittle behavior. The microsized epoxy specimens had a plastic deformation behavior resulting in a substantially higher ultimate tensile strength (up to 380 MPa) and strain at break (up to 130 %) compared to their bulk counterpart (68 MPa and 8%). Polarized light microscopy confirmed a rearrangement of the internal epoxy network structure during loading, resulting in the plastic deformation of the microscale epoxy. This was further accompanied by in-situ electron microscopy to further determine the deformation behavior of the micro-specimens during tensile loading and make accurate strain measurements using video-extensometry. This work thus provides novel insights on the epoxy material behavior at the confined microscale as present in fiber reinforced composite materials.

## 1. Introduction

Fiber-reinforced polymer composites are one of the go-to materials for high-end applications with excellent mechanical properties at low mass, used in the aerospace, wind energy, and automotive sectors [1,2]. Their inner structure consists of individual fibers, with a typical diameter of 5 µm (carbon fibers) to 20 µm (glass fibers), surrounded by a matrix material such as an epoxy resin. The fibers are usually stacked as textile layers on top of each other, creating a laminated microstructure as visible in Figure 1. This results in zones of interconnected matrix materials that have a dimension in the (sub)micron range such as the resin rich zones between fibers (1–25 µm, “intralayer”), as well as resin-rich zones in between reinforcement layers (10–200 µm, “interlayer”).

Several researchers pointed out that these microscale resin regions may have different mechanical properties than their macroscopic/bulk counterpart [3,4,5,6,7]. This has important consequences when it comes to understanding the deformation behavior of fiber reinforced polymer composites. For example, microscale modelling of composites [8,9,10,11,12,13,14] is done very frequently to predict and optimize a composite material without the requirement of expensive and time-consuming experimental trial-and-error. Yet, this requires an accurate input of the matrix material behavior at the microscale, where it is already known that bulk mechanical properties do not result in sufficient agreement between the models and reality [15]. Understanding the micromechanical properties and deformation behavior of the microconstituents is thus very important in order to increase and optimally engineer composite materials. Nevertheless, the characterization of matrix materials such as epoxy resins is typically performed at the bulk scale, using the prescribed standardized test methods such as ASTM D638 [16] or ISO 527 [17]. The research into microscale (epoxy) matrix properties remains limited [3,4,5,7,18,19] although these microscale matrix properties are likely more appropriate to represent the behavior of the matrix material confined between reinforcing fibers in actual composites.

The influence of specimen size on the mechanical properties is already well known for many materials like metals and ceramics [18,20,21,22,23,24]. Yet, for thermoset materials such as epoxies only a handful of studies are available [4,5,6,7]. In this respect, a better understanding of epoxy at the microscale is of utmost importance as it is the most used matrix material in composites. Odom and Adams were the first to describe a size effect in an epoxy material [5]. They observed that smaller cross-sections led to an increase in ultimate tensile strength (from 44 ± 11 to 94 ± 18 MPa for a decrease in cross-sectional area from 90.17 to 1.25 mm^2^). They concluded that the decrease in volume was the reason for this increase in strength since the flaw size decreased with smaller cross-sections. Yet, these volumes are still much larger than those present between the fibers of a composite. Da Silva et al. observed a similar effect for the mode II toughness in adhesion layers [25].

Hobbiebrunken et al. introduced a new production method which enabled them to make microfibers from epoxy resin that have a similar volume and shape as the volumes present in composite materials (fiber diameters between 22 and 52 µm) [4]. Such microfibers indeed offer a good model for the pillar like microscale matrix zones in between the reinforcing fibers. Indeed, the structure of the matrix material in a unidirectional fiber reinforced composite resembles a geometry of interconnected pillars with diameters in the micrometer region and a much longer length. These microfibers can then be analyzed using regular fiber tensile testing techniques. An increase in ultimate tensile strength showed that microscale fibers have a higher strength than larger bulk scale specimens. Misumi et al. expanded the research by studying the yield strength and stiffness in addition to the ultimate tensile strength for five different epoxy systems [6]. They observed an increase in both yield and ultimate tensile strength for all systems. The stiffness seemed to be unaffected by reducing the volume. Recently, Sui et al. conducted a study over a broader range of microfiber diameters to compare the mechanical properties at the microscale with those at the bulk scale [7]. Again a decrease in diameter resulted in an increase in ultimate strength and strain, as supported by the observations of Misumi et al. and Hobbiebrunken et al. [4,6]. In contrast to Misumi et al., however, Sui et al. reported an increase in stiffness for microsized specimens. Both Misumi et al. and Sui et al. reported comparable moduli for their tested microscale epoxy fibers of around 3 GPa, which is a typical value for bulk epoxy [6]. However, Sui et al. reported a modulus of only 1 GPa for the bulk material, which is quite low for an epoxy resin. This raises the question of whether the reported modulus increase was effectively induced by the microvolumes or by experimental deviations (e.g., differences in conversion, different testing method, etc.)

Overall, it is clear that the mechanical behavior of epoxies may change depending on its size. However, there is currently no agreement whether these changes are induced by a size-effect (related to the amount of defects inside the material), by a difference in microstructure (e.g., orientation of polymer chains), and/or by the production method of the specimens (e.g., a difference in conversion degree). While several researchers describe necking and ductility of the epoxy microfibers, only post-mortem observations are currently available. Furthermore, the number of tested samples is limited and there is a large variability reported in the results, even more so as strain measurement (clamp displacements) and strain and stress definition (nominal or true) are not always clearly mentioned.

Therefore, we investigate the microscale deformation behavior of epoxy resin, using microfiber shaped specimens similar to Ref. [4]. We combine tensile testing, thermal analysis, and in-situ electron microscopy, allowing for a deeper analysis of the epoxy behavior. First, via the measurement of the glass transition temperature we study possible differences in conversion and internal structure due to the production process which is mainly neglected in previous papers. To observe the necking phenomena as described in previous research, normal mechanical testing is combined with in-situ scanning electron microscopy (SEM) with the goal to provide more insights in the necking behavior. In addition, polarized light optical microscopy enables us to study possible rearrangements in the internal network structure of the microscale epoxy samples due to deformation since this was only briefly discussed up till now. A wide range of diameters are analyzed to observe any trends related to specimen size.

## 2. Materials and Methods

### 2.1. Materials

The epoxy resin used was Epikote^TM^ Resin MGS RIMR 135, which is based on diglycidyl ether of bisphenol A (DGEBA), combined with a liquid diamine hardener, Epicure^TM^ Curing Agent MGS RIMH 137 (Momentive Specialty Chemicals, Hemiksem, Belgium). Both components were mixed in a 100:30 weight ratio and degassed in a vacuum desiccator for 10 min. After optimal curing, 24 h at room temperature followed by 15 h at 80 °C, this leads to an epoxy system with a *T*_g_ between 80 and 90 °C. This curing sequence is used for the production of wind turbine blades, making it a relevant process to study.

### 2.2. Methods

#### 2.2.1. Production

Fine epoxy fibers were made via an optimized method based on the work of Hobbiebrunken et al. and further adjusted by Fiedler et al. [4,18] by drawing fibers from a vitrifying degassed epoxy mixture. For the specific system used in this paper, the resin mixture was first cured for 24 h at room temperature in an acclimatized room. This was followed by heating up the resin to 80 °C. After 7 min at 80 °C, the epoxy network developed sufficiently and allowed the drawing fibers out of the resin without breaking them. Fibers were drawn from the resin using an in-house developed automated dip-coater equipped with a needle (B. Braun Sterican stainless steel needles, beveled, 0.8 mm diameter, Diegem, Belgium) (Figure 2a). The needle was submerged into and immediately pulled out of the epoxy resin at a constant speed of 10 mm/s, drawing the epoxy into a microfiber. The produced fibers were transferred to a rack, both ends connected to the rack, and put in an electrically heated oven at 80°C for 15 h to complete the curing cycle (Figure 2b). Once the second heating cycle has ended, the fibers were taken from the rack and glued onto paper supports with a two-component 5-min epoxy glue (Figure 2c). These fiber supports are based on the ISO 11566 standard for the determination of the tensile properties of single fibers [26]. A lower diameter limit of around 20 µm was observed for this production method since thinner fibers break during production or become too difficult to handle [3,6,7,27].

In total, 42 microfibers were produced with a diameter ranging from 30 to 400 µm and a standard deviation of less than 10%. The variation in diameter was attributed to a variation in submerging depth of the needle and reaction time of the epoxy mixture. Figure 2d depicts cross-sections of several produced fibers after fracture, showing that the microfibers had a round cross-section, a smooth surface, and no visible porosities or defects.

Both bulk and microscaled epoxy specimens were produced using the same curing conditions to minimize any change in properties induced by a difference in conversion. This was confirmed by Modulated Differential Scanning Calorimetry (TA Instruments Q2000 DSC, Antwerp, Belgium), heating rate of 3 °C/min, modulation of 0.5 °C/min, scan from 0 to 150 °C. The glass transition temperature *T*_g_ of bulk scale and microscale specimens was 86.8 ± 1 °C and 87.3 ± 3 °C respectively. As the *T*_g_ is directly linked to the conversion degree [28], this shows that both bulk and microscale specimens have a similar conversion degree. In addition, no additional exothermic peak was measured above the *T*_g_ for both types of specimens.

#### 2.2.2. Mechanical Tests

The tensile properties of the fibers were determined through tensile tests that agreed both with the ISO 527-1 (strain rate) and ISO 11566 standard (specimen geometry, strain rate), for bulk and fiber materials, respectively [17,26]. A dedicated single fiber testing machine (Textechno Favimat, force resolution 0.01 mN, Mönchengladbach, Germany) was used, which allowed sufficient force and displacement resolution for testing microfiber specimens. The paper holder with a fiber was placed between the clamps at the glued points, the gauge length is 10 mm for all tests. Once fixed in the correct position, the paper support was cut in half and the test was started. A constant strain rate of 10%/min was selected to exclude strain rate effects during the comparison of the results. Since necking occurred in the fibers, it was not possible to obtain correct true stress—the true strain curves of the full tensile test. Therefore, force-displacement curves were selected to represent the deformation behavior of the microfibers. The diameter at fracture was determined post-mortem and used to calculate the ultimate tensile stress in order to obtain the true stress value at break. Mechanical data for the bulk scale were reused from previous research obtained in our group as published by Allaer et al. [29].

#### 2.2.3. Microscopical Observations

An Olympus BX-51 optical microscope with polarized light (Olympus, Antwerp, Belgium) was used to inspect all fibers before and after testing. At least 5 pictures were used to measure the diameter of the fibers over the complete gauge length via the online available Fiji software [30]. For each picture 5 measurements of the fiber diameter were done. The smallest measurement of the diameter was then used to calculate the cross-sectional area since the fiber will experience the highest stress at that point. After testing, the final diameter at the failure point was measured again and used to determine the true stress at break.

In addition to the tensile experiments above, several tensile tests were performed in a Scanning Electron Microscope (SEM, Thermo Fisher Scientific Phenom XL, Eindhoven, Netherlands). Here, a dedicated tensile stage was used (Tensile Sample Holder for Phenom XL, 150N load cell, Eindhoven, Netherlands) that could be controlled during SEM imaging. As the accuracy of the load cell was within the range of loads exerted by the fibers, these tests were used to obtain quantitative information about the deformation behavior during tensile loading, i.e., displacement and strain (not stress). A step-and-shoot method at preselected strain levels was applied to achieve high-quality images.

## 3. Results and Discussion

### 3.1. Mechanical Comparison between Bulk and Microscale Epoxy

Mechanical analysis of the microfiber epoxy samples revealed a high overall ductility with nominal strains reaching over 100%, in contrast to bulk epoxy samples with the same conversion rate. A representative force-displacement curve for a microfiber epoxy sample, Figure 3a, illustrates linear elasticity (zone I) followed by plastic deformation (zone II). For bulk scale samples this almost immediately leads to fracture, as only minor plastic deformation before failure is encountered for the majority of epoxy systems used in composites. On the other hand, the microfibers sustain this local decrease in diameter leading to a stable necking process (Figure 3b) and fracture is only encountered in a much later stage.

The necking proceeds throughout the whole microfiber resulting in a large plateau in the force- displacement curve; this plateau covers around 50% of nominal strain (due to the presence of the necking region, the tensile strain differs locally), resulting in an extreme ductility of the microfiber specimens (zone III, drawing of the microfibers). Finally, an increase in stress is observed (zone IV) and brittle fracture occurs. Indeed, the cross-sections shown in Figure 2, obtained after tensile fracture, show almost no plastic deformation at the fracture surface. The increase in stress is likely related to strain hardening occurring during the necking of the microfiber [7,31].

The obtained mechanical data shows that the microfibers generally have a lower yield stress compared to the bulk samples (Figure 3c, the engineering stress–engineering strain curves are representative up to the yield point as the lateral contraction is still limited and thus the initial cross-sectional area can be used). An increase in E-modulus can be observed with decreasing microfiber diameter, the smaller the fibers become the stiffer they are (Figure 4a). This is probably due to a more perfect network structure with very few or no defects. Yet, the presence of small defects will have a large influence on the fibers resulting in more variability of the reported values compared to their bulk counterparts. This has also been reported by other researchers [6,7] and is likely attributed to the fact that at these small scales any defects present in the specimens will have a bigger influence on the mechanical properties (e.g., premature failure) than in bulk-sized specimens.

It must be noted that for the microscale tests, the crosshead displacement of the tensile stage was used to determine the strain. This likely leads to an underestimation of the E-modulus since any fixture compliance (e.g., slippage, backlash) results in an artificially increased strain. It can thus explain why the bulk scale modulus, which was accurately measured through DIC and clip-on extensometers, is higher than those of the microfibers. Indeed, one would expect that the more perfected network structure of the microfibers would result in a higher stiffness. This strain measurement aspect is barely addressed in published literature on epoxy microfibers, yet the results clearly show the need for (video-)extensometry for these small scale specimens. It is suggested that using the in-situ measured specimen strain, the trend of increasing stiffness for smaller diameter specimens remains as observed in Figure 4a yet shifts to a slightly higher absolute value. This is further discussed in Section 3.2.

The microfibers show a similar yielding behavior over the full tested diameter range. In general, the observed yield stress is somewhat lower than the values obtained for the bulk sample, while the yield strain is similar to the bulk values (Figure 4c,d). Again, the latter can likely be attributed to the strain being based on crosshead displacements.

After yielding, the microfibers start to deform plastically resulting in necking. Similar to the yield stress, the necking stress and necking strain remain relatively unaffected by the diameter range tested here. The necking stress of the microfibers averages around 55–60 MPa (Figure 4e) while the necking strain averages around 4.5% of strain (Figure 4f). This indicates that for the microscale specimens there is somewhat less restriction for the epoxy network to deform and network chains possibly slip over one another resulting in reorientation of the chains in the internal network [7,31,32], somewhat similar to the deformation behavior of thermoplastics [33]. This is in line with the Wide Angle X-ray Scattering results of Sui et al. which showed that the necked region in epoxy microfibers had preferred internal orientations [7]. Moreover, this necking phenomenon of microscaled epoxy specimens has also been reported for other epoxy systems, indicating that this behavior might be universally valid for these thermoset materials [4,6,7].

Figure 5a schematically illustrates the change in network structure during the deformation of the microfiber specimens, resulting in an oriented network structure after necking. Initially, the epoxy network is similar to that of a bulk material and can be considered random. During necking, the network has the possibility to align itself in the tensile direction by sliding and reorientation of the network chain segments, explaining the sudden and relatively large decrease in cross-sectional area. This reorientation is indeed confirmed by the sharp increase in stress once the necking has proceeded completely through the gauge area of the fiber. (Figure 3a, zone IV). This (local) strain hardening effect can be explained by a more oriented state of the molecular network. After this reorientation, the stress increases and the epoxy network finally fractures via the rupture of chains (chain scission) as it cannot accommodate the increasing tensile strain anymore.

Polarized optical microscopy of tested fibers further confirms the change in orientation of the molecular network in the necked zone (Figure 5b). Starting at the neck, the oriented network interacts with the polarized light resulting in a bright zone. In contrast, the unnecked zone barely interacts with the light, resulting in a low amount of light passing through the second polarizer and thus a dark zone. This confirms that a random network is present with no preference for any direction before necking, while a reorientation at molecular level occurs during necking.

The necking and rearrangement of the network structure is accompanied with a drastic increase in ultimate tensile strength, with values ranging from 2 up to 6 times that of the bulk material (Figure 4g, the true stress values are calculated using the actual fiber diameter measured after fracture). Moreover, an increasing trend in the ultimate tensile stress is observed for decreasing fiber diameters (Figure 4h). Whereas the fibers with a diameter around 300 µm have an ultimate tensile strength in the range of 100–200 MPa, the finest fibers (around 50 µm) go up to 380 MPa (note that the bulk resin has ultimate tensile strength of only 68 MPa). Similar to the ultimate tensile strength, the strain at break increases drastically for all of the microfibers, with values going up to 130% (the bulk material has a strain at break of 8%).

Not all fibers reach these very high ultimate tensile strains as some fracture earlier during the necking process (zone III) which may be attributed to the presence of voids or defects. Due to the small volume, the chance of a void or defect being present is low which results in higher possible strengths [4,5]. Yet, if a void is present this will largely influence the mechanical behavior of the microfiber, resulting for example in early failure [21,34].

### 3.2. In-Situ Scanning Electron Microscopy Inspection of the Tensile Deformation of Epoxy Microfibers

To have a better understanding of the necking behavior a set of microfibers were mechanically tested with in-situ scanning electron microscopy during the necking process. To obtain the best possible quality of pictures, a step-and-shoot method was applied at preselected displacement intervals. Via these pictures, the diameter of the fiber is accurately measured which provides the possibility to calculate the change in diameter due to necking and the corresponding change in length (Equation (1)). Assuming a constant volume (ν = 0.5, plasticity), the strain in the fiber direction can be calculated from the diameter variation via Equation (1):(1)$$Tensile\;strain=\frac{\left(D_{1}^{2}-D_{2}^{2}\right)}{D_{2}^{2}}$$
where D1 and D2 are the diameter just in front of the neck and just behind the neck (Figure 6).

The necking process that takes place during the plastic deformation of the microfibers can be clearly seen in Figure 6. The measured diameter contraction after necking and corresponding tensile strain (Equation (1)) for several microfibers are listed in Table 1. All tested microfibers showed a decrease in diameter between 16% and 20%, with an average value of 18.3%, corresponding to a strain in the fiber direction ranging from 40% to almost 60% (average of 49.9%). Note the measured contraction is only due to the necking process, the tensile strain developed in the linear elastic zone or after necking of the fiber is not considered here.

In addition to calculating the strain according to Equation (1), the idea of video-extensometry is proposed as well to directly measure the increase in tensile strain after necking. We therefore deliberately handled a couple of specimens without gloves, resulting in small contamination on the surface of the specimens that are ideal as surface markers, see Figure 7. These surface markers did not affect the mechanical properties of the fibers and allowed to directly calculate the tensile strain by the change in distance between the markers. The results are listed Table 2 and confirm that the values obtained through Equation (1) are correct.

The use of surface markers is indeed interesting for quantifying the local deformation behavior of materials. Not only do these offer accurate strain data as an alternative to the nominal displacement measured at the clamp jaws, this could also lead to a more accurate determination of the Young’s modulus as well as a true stress–true strain curve for small scale specimens, similar to the use of extensometers or strain gauges for bulk specimens.

Yet, to enable accurate measurement of the Young’s modulus a more optimized marker pattern needs to be applied. The displacements in the linear elastic zone are much smaller than those observed during plastic deformation (Figure 7 top left two images). The Young’s modulus is typically measured over a strain range of 0.01 which means that over a visible specimen length of 200 µm a displacement of only 2 µm will be observed. This fell within the noise level of the SEM images, resulting in unreliable values of Young’s modulus with the above used method. Therefore, future work will focus on optimizing the type of markers or by creating a random and dense pattern of marker particles that enables Digital Image Correlation (DIC) on the SEM images to reach a higher resolution of the displacement measurements.

## 4. Conclusions

A set of microfibers was produced and studied via standard tensile testing. The deformation behavior of the confined microfiber epoxy samples was substantially different compared to bulk epoxy behavior, although thermal analysis confirmed a similar conversion rate of the epoxy in both the microfiber and the bulk state. During both linear elastic deformation and yielding, the microfibers behaved very similarly to the bulk samples. Yet, after necking, the behavior of the microscale and the bulk scale samples is very different. Indeed, the bulk scale samples show fracture shortly after yielding, whereas the microfiber specimens showed a large plastic deformation via the formation of a necked zone that extended throughout the fiber. As a result, much higher strains at break (up to 130% vs. 8% for bulk) and ultimate tensile strengths (up to 380 MPa vs. 68 MPa for bulk) were observed for these confined microfibers.

This high plastic deformation is attributed to a rearrangement of the epoxy network structure in the microfibers due to chain slippage and some orientation along the tensile loading direction, as confirmed by polarized optical microscopy of the plastically deformed microspecimen and the occurrence of strain hardening.

In-situ electron microscopy, moreover, allowed us to accurately observe the deformation behavior of these microfibers and measure tensile strains and diameter contractions during loading. An increase in length of around 50% was observed due to necking only.

The remarkable difference between the deformation behavior of epoxy resin at the micro- and macroscale clearly illustrates the need of microscale testing. Indeed, the resin pockets in fiber reinforced polymer composites are confined microsized regions, which cannot be accurately characterized via bulk testing. Especially for predictive modelling tools, such as micromechanical modelling, this necessitates the use of microscale measured properties to obtain correct data and insights.

## Figures and Tables

**Figure 1 polymers-12-02581-f001:**
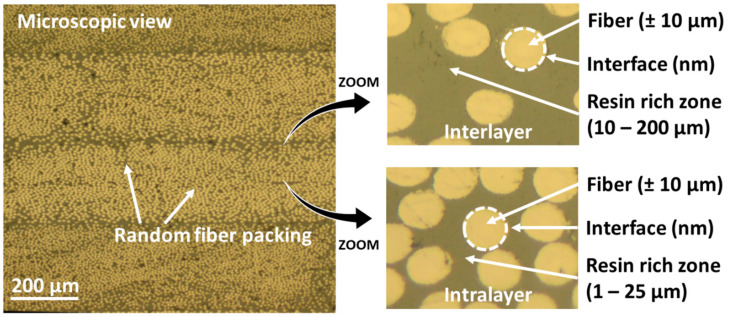
Overview of the different scales present in a fiber reinforced composite. The actual size of epoxy material domains in composites is in the range of 1–200 µm.

**Figure 2 polymers-12-02581-f002:**
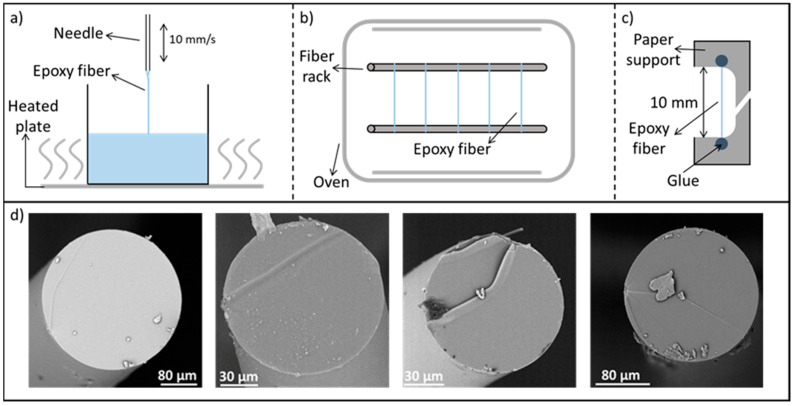
Production method. (**a**) Epoxy fibers are drawn from the vitrifying resin. (**b**) The produced fibers are cured at 80 °C in an oven for 15h. (**c**) The final fibers are glued on paper supports for single fiber tensile testing. (**d**) Cross-sections of several produced microfibers after fracture, showing a perfect circular shape.

**Figure 3 polymers-12-02581-f003:**
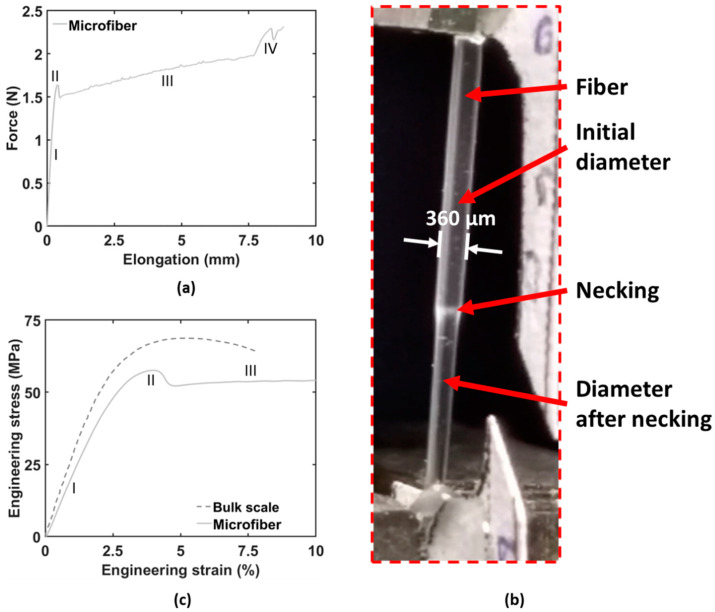
(**a**) Representative force—displacement curve of a microfiber. Four zones can be observed: I—linear elastic zone, II—necking formation III—constant necking IV - strain hardening. Nominal strain is based on clamp displacement. (**b**) Close-up of an epoxy microfiber during tensile testing, the necking of the microfiber can be clearly seen. A large change in diameter is present between the necked zone and the original microfiber. (**c**) Stress–strain curve of a microfiber and bulk scale (the latter adapted from previous work. [29]).

**Figure 4 polymers-12-02581-f004:**
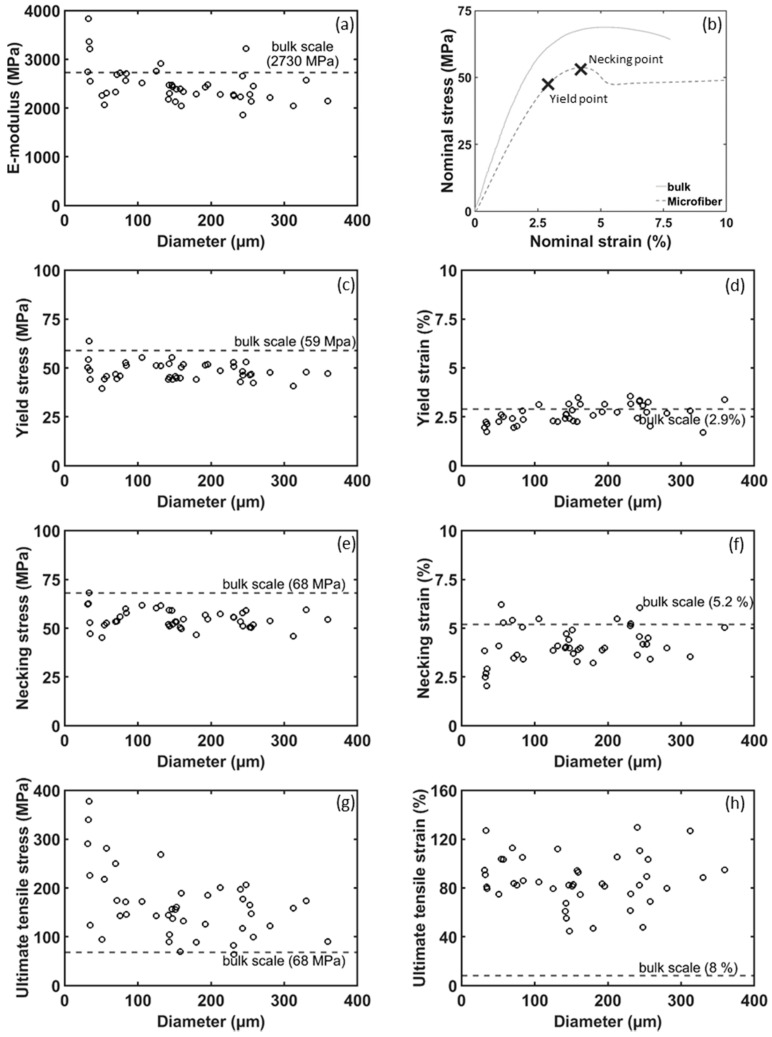
Representation of all mechanical data (**a**) E-modulus (**b**) engineering stress–engineering strain curve of representative microfiber tensile test with indication of necking and yield point (**c**) yield stress (**d**) yield strain (**e**) necking stress (**f**) necking strain (**g**) ultimate tensile stress and (**h**) ultimate tensile strain.

**Figure 5 polymers-12-02581-f005:**
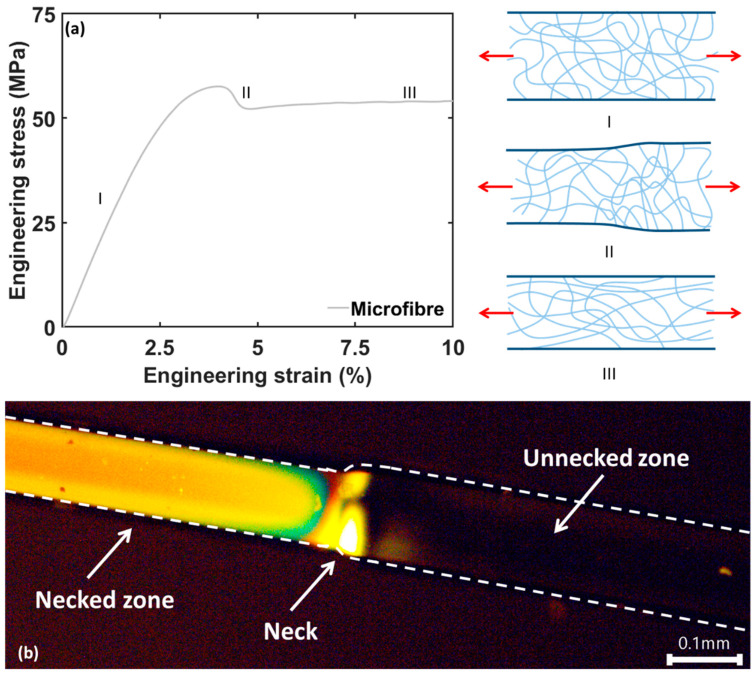
Schematic representation of the rearrangement of the epoxy network under tensile stress (**a**). Polarized optical microscopy confirms an oriented molecular microstructure in the necked zone of the specimens (**b**).

**Figure 6 polymers-12-02581-f006:**
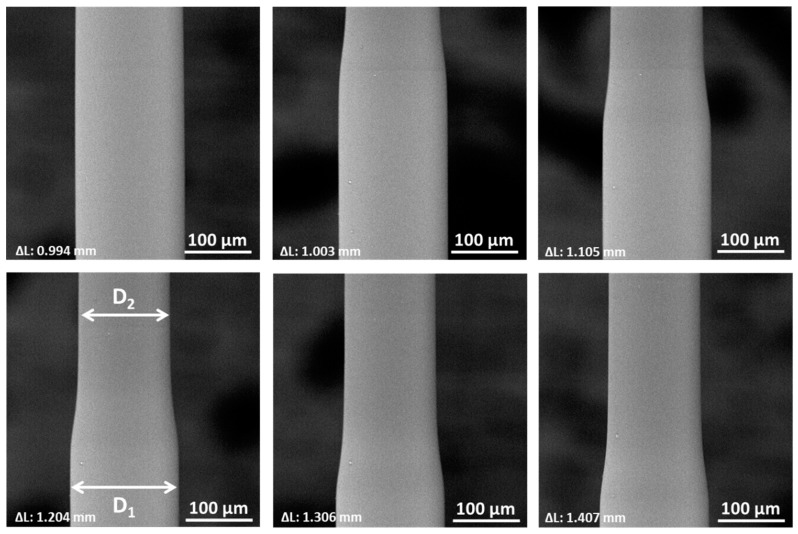
Sequential SEM images of necking in a microfiber. ∆L indicates the displacement of the clamps.

**Figure 7 polymers-12-02581-f007:**
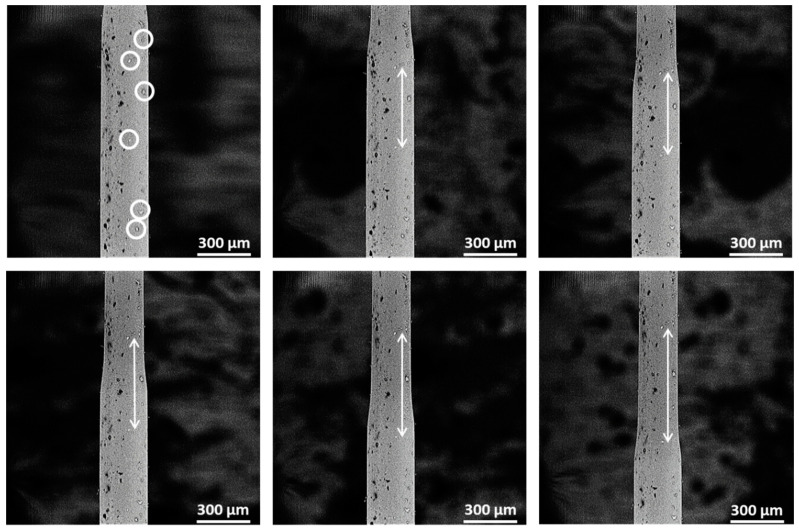
Microfiber with several particles on the surface for tracking the elongation. White arrows show the change in length during the necking propagation.

**Table 1 polymers-12-02581-t001:** Calculation of the diameter contraction of several microfibers and the theoretical calculation of the strain due to the necking according to Equation (1).

	Diameter Prior to Necking (µm)	Diameter Just after Necking (µm)	Diameter Contraction (%)	Strain (%)
Microfiber 1	21.3	17.2	19.3	53.5
Microfiber 2	113.0	90.1	20.3	57.3
Microfiber 3	141.4	117.6	16.8	44.5
Microfiber 4	154.2	126.0	18.3	49.8
Microfiber 5	168.1	136.8	18.6	51.0
Microfiber 6	275.7	225.0	18.4	50.1
Microfiber 7	275.8	223.6	18.9	52.1
Microfiber 8	424.7	347.7	18.1	49.2
Microfiber 9	610.0	412.0	16.0	41.8

**Table 2 polymers-12-02581-t002:** Comparison of the strain calculated based on the diameter reduction and the displacement of surface markers.

	Calculated Strain via Diameter (%)	Strain (%)
Microfiber 6	50.1	52.5
Microfiber 7	52.1	53.7
Microfiber 9	41.8	41.0

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
