# Peer review of "In-Situ Observations of Microscale Ductility in a Quasi-Brittle Bulk Scale Epoxy"

_polymers, 2020, doi:10.3390/polym12112581_

Round 1

Reviewer 1 Report

See the attached document, which is the review for the Authors.

Reviewer 2 Report

In this manuscript, a set of epoxy microfibers were prepared and studied via standard tensile testing, polarized light microscopy, in-situ electron microscopy and thermal analysis. The results show the great differences between the deformation characteristics of epoxy samples in the microfiber and bulk states. This work can offer novel insights on the micromechanical behavior of epoxy materials which are a surrounding matrix in fiber reinforced composite materials. This research subject is of great significance. This manuscript is well organized, the important viewpoints are clearly presented, and the conclusions are supported by the data. The research subject accords with the scope of Polymers. This manuscript can be accepted for publishing.

Please note the following minor issues as authors revise the manuscript:

Abstract

I suggest authors to re-organize two following sentences in order to let readers to understand them: The aim is to have a high volume fraction of fibers well-dispersed in the matrix material. As a result, the matrix phase in unidirectional composites is composed of many interconnected, microscale (1–200 μm), tube-like volumes confined by the reinforcing aligned fiber bundles.

Lines 114-120: English tenses in Lines 114-120 should be the same as those in Lines 109-114.

Line 148: “Mechanical data for the bulk scale was reused from ---”: “was” should be replaced by “were”.

Lines 153-154: “The thinnest point was used for the calculations since this point will experience the highest stresses”: I suggest authors to improve “this point will experience the highest stresses”.

Lines 214-215: “The latter can likely again be attributed due to the strain being based on crosshead displacements”: “be attributed due to” should be replaced by “be attributed to”.

Line 250: “Polarized optical microscopy of tested fibers further confirm the change in”: “confirm” should be replaced by “confirms”.

Line 261, Line 308: “a set of microfibers was” should be replaced by “a set of microfibers were”.

Line 268: “Where” should be replaced by “where”.

Round 2

Reviewer 1 Report

See the attached document, which is the review for the Authors.

Author Response

We are glad that the reviewer recognizes the importance of our work for the field of fibre reinforced polymer composites. Many of the comments of the reviewer were similar to the first review round and we would like to refer to our previous rebuttal document for our response. We still believe there might have been a mistake in trying to replicate our results as we investigate pure epoxy resin in the form of microfibers and not a fibre reinforced composite (referring to the reviewer's statement about fibres shifting in a matrix).